# *Capsicum annuum* with causal allele of hybrid weakness is prevalent in Asia

**Kumpei Shiragaki[1¤], Shonosuke Seko[2], Shuji Yokoi[1,2,3,4], Takahiro Tezuka[1,2,3]***

**1** Graduate School of Life and Environmental Sciences, Osaka Prefecture University, Sakai, Osaka, Japan, **2** Graduate School of Agriculture, Osaka Metropolitan University, Sakai, Osaka, Japan, **3** Education and Research Field, School of Agriculture, Osaka Metropolitan University, Sakai, Osaka, Japan, **4** Bioeconomy Research Institute, Research Center for the 21st Century, Osaka Metropolitan University, Osaka, Japan

¤ Current address: Graduate School of Agricultural and Life Sciences, University of Tokyo, Tokyo, Japan
* tezuka@omu.ac.jp

**Data Availability Statement:** All relevant data are within the paper and its Supporting Information files.

**Funding:** This work was partly supported by JSPS KAKENHI Grant Numbers JP17K15224 and

## Abstract

Reproductive isolation, including hybrid weakness, plays an important role in the formation of species. Hybrid weakness in *Capsicum*, the cessation of plant growth, is caused by two complementary dominant genes, *A* from *C. chinense* or *C. frutescens* and *B* from *C. annuum*. In the present study, we surveyed whether 94 *C. annuum* accessions had *B* or *b* alleles by crossing with *C. chinense* having the *A* allele. Of the 94 *C. annuum* accessions, five had the *B* allele, three of which were native to Latin America and two were native to Asia. When combined with previous studies, the percentage of *B* carriers was 41% in Japan, 13% in Asia excluding Japan, 6% in Latin America, and 0% in Europe and Africa. In addition, 48 accessions of *C. annuum* from various countries were subjected to SSR analysis. Clades with high percentages of *B*-carriers were formed in the phylogenetic trees. In the principal coordinate analysis, most *B*-carriers were localized in a single group, although the group also included *b*-carriers. Based on these results, we presumed that the *B* allele was acquired in some *C. annuum* lines in Latin America, and *B*-carriers were introduced to the world during the Age of Discovery, as along with the *b*-carriers.

## Introduction

Cross-breeding is commonly used in plant breeding to introduce new desirable traits from related species into cultivated species. However, reproductive isolation mechanisms, which are divided into prezygotic and postzygotic isolation, often prevent inter- or intra-specific crosses. Postzygotic isolation is further divided into several phenomena including seed abortion, hybrid weakness, and hybrid breakdown. Hybrid weakness is defined as the weak growth of $F_1$ hybrids from parents showing normal growth and has been reported in several plant species [1–4]. Hybrid weakness is also called hybrid lethality or hybrid necrosis, depending on symptoms or species. In most cases, the interaction of two complementary genes causes hybrid weakness, which is consistent with the Bateson-Dobzhanzky-Muller model, and autoimmune responses are involved in these phenomena [1, 4–6].

The *Capsicum* genus comprises approximately 35 species, including five cultivated species: *C. annuum*, *C. chinense*, *C. frutescens*, *C. baccatum*, and *C. pubescens*. Although *C.*

JP20K05988 from the Japan Society for the Promotion of Science (to TT) (https://www.jsps.go.jp/english/index.html). The funders had no role in study design, data collection and analysis, decision to publish, or preparation of the manuscript.

**Competing interests:** The authors have declared that no competing interests exist.

*annuum* is the most common commercial species, some useful traits (i.e., abiotic and biotic stress tolerance [7, 8] and multiple flowers [9]) have been found in *C. chinense* and *C. frutescens*, which are related to *C. annuum*. It is expected that these traits can be introduced into *C. annuum* by cross breeding. However, reproductive isolation often works between *C. annuum* and *C. chinense* or *C. frutescens* [10–15]. Therefore, understanding the mechanism and developing methods to overcome reproductive isolation are needed for cross-breeding in *Capsicum*.

*C. annuum* is native to Latin America and has been introduced worldwide since Christopher Columbus discovered the West Indies [16]. During the 16–17th centuries, *Capsicum* migrated to Europe and Africa from Latin America, and to Asia by Portuguese [16]. During the 17–18th centuries, *Capsicum* from Latin America was brought several times to Europe, the Middle East, and Asia by trade [16]. Additionally, it has been suggested that *Capsicum* has evolved in each country where *Capsicum* was introduced [17–19], although the evidence is weak at present.

Hybrid weakness, formerly called as "hybrid dwarfism,", is observed in some reciprocal crosses of *C. annuum* × *C. chinense* or *C. annuum* × *C. frutescens* [11, 12]. $F_1$ hybrids having hybrid weakness show cessation of growth approximately 40 days after germination and a hypersensitive response-like reaction, including $H_2O_2$ accumulation, programmed cell death, and upregulation of defense-related genes in leaves [20]. *Capsicum* hybrid weakness is caused by two complementary dominant genes, and it has been reported that *C. annuum* can exhibit the *aaBB* genotype, whereas *C. chinense* and *C. frutescens* exhibit only the *AAbb* genotype. However, extensive studies have not been conducted on *C. chinense* and *C. frutescens* [11, 13]. Yazawa et al. [11] reported that the distribution of *C. annuum* cultivars with the *B* allele prevails in East Asia. However, the *C. annuum* cultivars used in the geographic survey of causal alleles were native to Asia and Latin America [11]; thus, their global geographic distribution is unknown.

In the present study, we report the geographic and phylogenetic distribution of *C. annuum* with the *B* allele of hybrid weakness using cultivars from across the globe to uncover the evolutionary history of the causal gene of hybrid weakness in *C. annuum*.

## Materials and methods

### Plant materials

We used 94 *C. annuum* accessions (including 16 *C. annuum* var. *glabriusculum* accessions; S1 Table), one *C. chinense* accession, one *C. frutescens* accession, one *C. baccatum* accession, and one *C. pubescens* accession (Table 1). After the seeds were germinated on moistened filter papers placed in Petri dishes, all seedlings were transplanted to pots (9 cm diameter, 10 cm depth) filled with culture soil (Sakata Super Mix A, Sakata Seed Co., Tokyo, Japan). The seedlings were cultivated in a constant-temperature room (25°C, 12 h light and 12 h dark, light intensity of 85 µmol $m^{-2}$ $s^{-1}$). At 30 days after germination, plants were transferred to bigger pots (21 cm diameter, 15 cm depth) and placed in a greenhouse (natural day length; Osaka Prefecture University, Sakai, Osaka, Japan) where the temperature was maintained above 15°C. For additional fertilization, plants were supplied weekly with Otsuka-A prescription (OAT Agrio Co., Ltd., Tokyo, Japan) containing 18.6 mM nitrogen, 5.1 mM phosphorus, 8.6 mM potassium, 8.2 mM calcium, and 0.4 mM magnesium. For the test cross, we used 94 *C. annuum* accessions (S1 Table) and *C. chinense* PI 159236 (genotype *AAbb*). For SSR analysis, we used 48 *C. annuum* accessions and one accession each of *C. chinense*, *C. frutescens*, *C. baccatum*, and *C. pubescens* (Table 1).

**Table 1. Plant materials used for SSR analysis.**

| Reference number [a] | Species | Accession | Cultivar name | Origin | *B* or *b* carrier |
|---|---|---|---|---|---|
| 1 | *C. annuum* var. *glabriusculum* | PI 631135 | Chiltepe | Guatemala | *b* carrier |
| 2 | *C. annuum* var. *glabriusculum* | PI 631137 | Tolito | Guatemala | *b* carrier |
| 3 | *C. annuum* var. *glabriusculum* | PI 311126 | | Nicaragua | *b* carrier |
| 4 | *C. annuum* var. *glabriusculum* | PI 439325 | | Nicaragua | *b* carrier |
| 5 | *C. annuum* | PI 201237 | Chilpolce Mico | Mexico | *b* carrier |
| 6 | *C. annuum* | PI 645486 | | Columbia | *b* carrier |
| 7 | *C. annuum* | PI 438536 | Chile verde | Belize | *b* carrier |
| 8 | *C. annuum* | PI 438538 | Dzabalan | Belize | *b* carrier |
| 9 | *C. annuum* | PI 209112 | | Puerto Rico | *b* carrier |
| 10 | *C. annuum* | PI 585246 | | Ecuador | *b* carrier |
| 11 | *C. annuum* | PI 241660 | | Peru | *b* carrier |
| 12 | *C. annuum* | PI 241646 | | Peru | *b* carrier |
| 13 | *C. annuum* | PI 213915 | | Bolivia | *B* carrier |
| 14 | *C. annuum* | PI 224413 | | Trinidad and Tobago | *B* carrier |
| 15 | *C. annuum* | PI 497971 | Ano Todo | Brazil | *B* carrier |
| 16 | *C. annuum* | PI 640449 | Szechwan 4 | Taiwan | *b* carrier |
| 17 | *C. annuum* | PI 640515 | Cheongryong | South Korea | *b* carrier |
| 18 | *C. annuum* | PI 286419 | | Nepal | *b* carrier |
| 19 | *C. annuum* | PI 645534 | | Papua New Guinea | *b* carrier |
| 20 | *C. annuum* | PI 640751 | | Vietnam | *b* carrier |
| 21 | *C. annuum* | PI 470243 | | Indonesia | *b* carrier |
| 22 | *C. annuum* | PI 127442 | | Afghanistan | *b* carrier |
| 23 | *C. annuum* | PI 135873 | | Pakistan | *b* carrier |
| 24 | *C. annuum* | PI 177301 | | Syria | *B* carrier |
| 25 | *C. annuum* | PI 181734 | | Lebanon | *b* carrier |
| 26 | *C. annuum* | PI 640562 | Kunja | South Korea | *B* carrier |
| 27 | *C. annuum* | Grif 9089 | | Greece | *b* carrier |
| 28 | *C. annuum* | PI 653659 | | Germany | *b* carrier |
| 29 | *C. annuum* | PI 439323 | | Netherlands | *b* carrier |
| 30 | *C. annuum* | PI 273415 | Long Hot | Italy | *b* carrier |
| 31 | *C. annuum* | PI 249908 | Pimentos Morrones | Portugal | *b* carrier |
| 32 | *C. annuum* | PI 645544 | | Tanzania | *b* carrier |
| 33 | *C. annuum* | PI 193471 | | Ethiopia | *b* carrier |
| 34 | *C. annuum* | PI 640770 | | Uganda | *b* carrier |
| 35 | *C. annuum* | JP 32511 | Sapporo Onaga Nanban | Japan | *B* carrier [11] |
| 36 | *C. annuum* | JP 32520 | Mie | Japan | *B* carrier [11] |
| 37 | *C. annuum* | JP 32523 | Akashi | Japan | *B* carrier [11] |
| 38 | *C. annuum* | JP 32549 | Yatsubusa | Japan | *B* carrier [11] |
| 39 | *C. annuum* | JP 32555 | Zairai | Japan | *b* carrier [11] |
| 40 | *C. annuum* | JP 32562 | Nikko | Japan | *b* carrier [11] |
| 41 | *C. annuum* | JP 32566 | Fushimiamanaga | Japan | *B* carrier [11] |
| 42 | *C. annuum* | JP 82498 | Takanotsume | Japan | *B* carrier [11] |
| 43 | *C. annuum* | JP 123787 | Shosuke | Japan | *B* carrier [11] |
| 44 | *C. annuum* | JP 124339 | Murasaki | Japan | *b* carrier [11] |
| 45 | *C. annuum* | PI 640723 | Shishito | Japan | *B* carrier [11] |
| 46 | *C. annuum* | | Miogi L3 | Japan | *b* carrier [13] |
| 47 | *C. annuum* | | Miogi L4 | Japan | *b* carrier [13] |

(*Continued*)

**Table 1.** (Continued)

| Reference number [a] | Species | Accession | Cultivar name | Origin | *B* or *b* carrier |
|---|---|---|---|---|---|
| 48 | *C. annuum* | | Enken-amanaga | Japan | *b* carrier [13] |
| 49 | *C. chinense* | PI 159236 | | USA | - |
| 50 | *C. frutescens* | PI 586675 | Tabasco | USA | - |
| 51 | *C. baccatum* | PI 640882 | | Peru | - |
| 52 | *C. pubescens* | PI 593624 | Chile de caballo | Guatemala | - |

[a] Reference number corresponds to the number in Fig 6

## Test cross

Conventional crossing was conducted as follows: flowers of *C. annuum* accessions (S1 Table) used as maternal parents were emasculated one day before anthesis and pollinated with the pollen of *C. chinense* PI 159236, used as paternal parents. $F_1$ seeds were collected from fully ripe fruits. After the $F_1$ seeds were germinated on moistened filter papers placed in Petri dishes, all seedlings were transplanted to pots (9 cm diameter, 10 cm depth) filled with culture soil (Sakata Super Mix A, Sakata Seed Co.). The seedlings were cultivated in a constant-temperature room (25°C, 12 h light and 12 h dark, 85 μmol m$^{-2}$ s$^{-1}$). At 60 days after germination, the genotypes of each accession were determined as the *aaBB* genotype if all $F_1$ hybrid plants did not obtain the first flower due to growth arrest, a characteristic of *Capsicum* hybrid weakness [20], and the *aabb* genotype if all $F_1$ hybrid plants received the first flower and continued to grow.

## DNA extraction

Total DNA of accessions for SSR analysis (Table 1) was extracted from individual leaves using the cetyltrimethylammonium bromide (CTAB) method [21] with minor modifications. Each leaf was ground in a mortar with liquid nitrogen. The ground leaf was mixed with CTAB isolation buffer (2% w/v CTAB, 1.4 M NaCl, 0.2% v/v β-mercaptoethanol, 20 mM EDTA, and 100 mM Tris-HCl, pH 8.0) preheated to 60°C, and the mixture was incubated at 60°C for 60 min. The suspension was extracted twice with chloroform/isoamyl alcohol (24:1) and centrifuged for 15 min at 500 *g*. The aqueous phase was transferred to a new tube, and nucleic acids were precipitated by the addition of isopropanol (2/3 volume) and centrifuged for 20 min at 1,000 *g*. The pellet was washed with 70% ethanol and dissolved in 50 μL of Tris-EDTA buffer (10 mM Tris-HCl and 1 mM EDTA, pH 8.0).

## SSR analysis

34 SSR markers were selected from Sugita et al. [22] (Table 2 and S2 Table). PCR was performed with a final volume of 10 μL, which contained 2 mM each of dNTP, 10× reaction buffer, 0.2 mM each of forward and reverse primer pairs, 0.5 U of Taq polymerase, and 50 ng of DNA as template. PCR reaction cycles consisted of initial denaturation at 94°C for 2 min, followed by 35 cycles of 94°C for 30 s, 54°C for 1 min, 72°C for 30 s, and final extension for 2 min at 72°C. The PCR products were separated by electrophoresis on 8% polyacrylamide gels in Tris-borate-EDTA. Electrophoresis was performed at 55 mA for 90 min at 25°C. The gels were stained with ethidium bromide and visualized using a UV lamp. Polymorphic alleles were scored as present or absent by visual inspection.

**Table 2. Polymorphic information of SSR markers in the present study.**

| SSR markers | Linkage group by Sugita et al [22] | Na | I | He | Ho |
|---|---|---|---|---|---|
| ge257-93pmA0742C | 1 | 4 | 0.873 | 0.474 | 0.000 |
| ge214-224pmc0769C | 2 | 5 | 1.194 | 0.607 | 0.000 |
| CAMS-358 | 3 | 4 | 0.879 | 0.463 | 0.000 |
| CAMS-861 | 4 | 4 | 0.464 | 0.215 | 0.000 |
| ge273-249pmc0783C | 5 | 3 | 0.634 | 0.385 | 0.000 |
| ge245-46pmA0870C | 6 | 7 | 1.705 | 0.785 | 0.000 |
| ge105-630pms0282C | 7 | 6 | 1.073 | 0.556 | 0.000 |
| Hpms 1–155 | 8 | 5 | 0.969 | 0.486 | 0.000 |
| ge247-859pmA1091C | 9 | 4 | 0.929 | 0.510 | 0.000 |
| ge54-242pmA0133C | 10 | 6 | 1.491 | 0.740 | 0.000 |
| Hpms 2–2 | 11 | 5 | 0.858 | 0.453 | 0.000 |
| CAMS-301 | 1 | 4 | 1.002 | 0.562 | 0.000 |
| es723TC3943S | 1 | 5 | 1.297 | 0.683 | 0.000 |
| ge232-524pmr0557C | 2 | 6 | 1.346 | 0.668 | 0.000 |
| CAMS-492 | 2 | 3 | 0.829 | 0.531 | 0.000 |
| es777CA516741S | 3 | 4 | 0.834 | 0.464 | 0.000 |
| ge279-375pmH0430W | 3 | 5 | 1.259 | 0.669 | 0.000 |
| es713BM064640S | 4 | 6 | 1.486 | 0.714 | 0.000 |
| PM12 | 4 | 6 | 1.343 | 0.679 | 0.000 |
| CAMS-051 | 5 | 5 | 1.287 | 0.674 | 0.000 |
| CAMS-020 | 5 | 2 | 0.540 | 0.355 | 0.000 |
| CAMS-361 | 6 | 3 | 1.008 | 0.603 | 0.000 |
| CAMS-351 | 6 | 5 | 1.221 | 0.624 | 0.000 |
| PM27 | 7 | 5 | 1.469 | 0.741 | 0.000 |
| CAMS-451 | 8 | 6 | 1.607 | 0.769 | 0.000 |
| CAMS-405 | 8 | 5 | 1.263 | 0.668 | 0.000 |
| CAMS-679 | 9 | 8 | 1.729 | 0.774 | 0.059 |
| CAMS-844 | 9 | 6 | 1.420 | 0.714 | 0.000 |
| CAMS-336 | 10 | 5 | 1.368 | 0.706 | 0.000 |
| PM18 | 10 | 5 | 1.247 | 0.660 | 0.000 |
| PM29 | 11 | 9 | 1.661 | 0.754 | 0.000 |
| PM37 | 11 | 7 | 1.584 | 0.749 | 0.000 |
| ge250-854pmr0454C | 12 | 3 | 0.810 | 0.479 | 0.000 |
| es743CA521534S | 12 | 5 | 1.292 | 0.672 | 0.000 |

Na, number of different alleles; I, Shannon's information index; He, Nei's unbiased gene diversity index; Ho, observed heterozygosity.

## Data analysis

GenAlEx 6.51 [23] was used to calculate the number of different alleles (Na), Shannon's information index, observed heterozygosity (Ho), and expected heterozygosity (He) for each SSR marker. Power Marker version 3.25 [24] was used to calculate genetic distance and phylogenetic construction, and MEGA X [25] was used for viewing the dendrogram. For the phylogenetic analysis, genetic distance was calculated using Nei's genetic distance [26], followed by phylogeny construction using the unweighted pair group method using arithmetic average (UPGMA) and neighbor-joining (NJ) methods. Branch support was assessed by bootstrap resampling with 1000 replicates. Principal coordinate analysis (PCoA) among populations was also conducted using GenAlEx 6.51 [27].

To infer the population structure of the pepper species used in this study, we used a model-based clustering algorithm implemented in the computer program Structure version 2.3.4 [28]. Analyses were based on an admixture ancestral model with correlated allele frequencies. The analysis was run for K = 1–10 with 10,000 Monte Carlo Markov Chain replicates after a burn-in of 10,000 replicates. For each value of K, 10 independent runs were performed to generate an estimate of the true number of subpopulations. The optimal K was selected using the Evanno method [29] estimated using Structure Harvester [30].

## Results

### Geographical distribution of hybrid weakness causal gene *B*-carrier in *C. annuum*

We surveyed whether 94 *C. annuum* accessions had the *B* allele of hybrid weakness by crossing with *C. chinense* having the *A* allele (S1 Table). Five accessions had *B* allele and the remaining 89 accessions had *b* allele. Among the *B*-carriers, three accessions were native to Latin America and two were native to Asia. The percentage of *B*-carriers was calculated for each geographic region (Latin America, Europe, Africa, Asia excluding Japan, and Japan), combining the present results with those of a previous study [11] (Tables 1 and 3, Fig 1). The percentage of *B*-carriers was 13% in all *C. annuum* accessions, 41% in those from Japan, 13% in those from Asia excluding Japan, 6% in those from Latin America, and 0% in those from Europe and Africa.

### Polymorphism and allelic diversity evaluated by SSR analysis

We analyzed 48 *C. annuum* accessions, including four *C. annuum* var. *glabriusculum* accessions (13 *B*-carriers and 35 *b*-carriers), one *C. chinense* accession, one *C. frutescens* accession, one *C. baccatum* accession, and one *C. pubescens* accession, using 34 SSR markers (Tables 1 and 2). The number of Na was 2–9, and the Shannon's information index of the genetic variation index was 0.464–1.729. The He was 0.215–0.785. The Ho value was 0.059 for CAMS-679 but 0 for the other markers.

### Genetic relationship analysis

Two phylogenetic trees were constructed using the UPGMA and NJ methods, based on analysis using 34 SSR markers (Figs 2 and 3). *C. annuum* accessions in Latin America, including *C. annuum* var. *glabriusculum* were close to those of other domestic species: *C. chinense*, *C. frutescens*, *C. baccatum*, and *C. pubescens* (Figs 2 and 3). *C. annuum* accessions did not form a region-specific clade on either UPGMA or NJ trees (Figs 2 and 3). The number of clades with a high percentage of *B*-carriers was two in the UPGMA tree and one in the NJ tree (Figs 2 and

**Table 3. Geographic distribution of *b* and *B* carriers.**

| Region | Yazawa et al. [11] | | | This study | | | Total | | |
| | *b* carrier | *B* carrier | Total | *b* carrier | *B* carrier | Total | *b* carrier | *B* carrier | Total |
|---|---|---|---|---|---|---|---|---|---|
| Latin America | 26 | 1 | 27 | 34 | 3 | 37 | 60 | 4 | 64 |
| Europe | - | - | - | 17 | 0 | 17 | 17 | 0 | 17 |
| Africa | - | - | - | 16 | 0 | 16 | 16 | 0 | 16 |
| Japan | 19 | 14 | 33 | 1 | 0 | 1 | 20 | 14 | 34 |
| Asia excluding Japan | 33 | 6 | 39 | 21 | 2 | 23 | 54 | 8 | 62 |
| All | 78 | 21 | 99 | 89 | 5 | 94 | 167 | 26 | 193 |

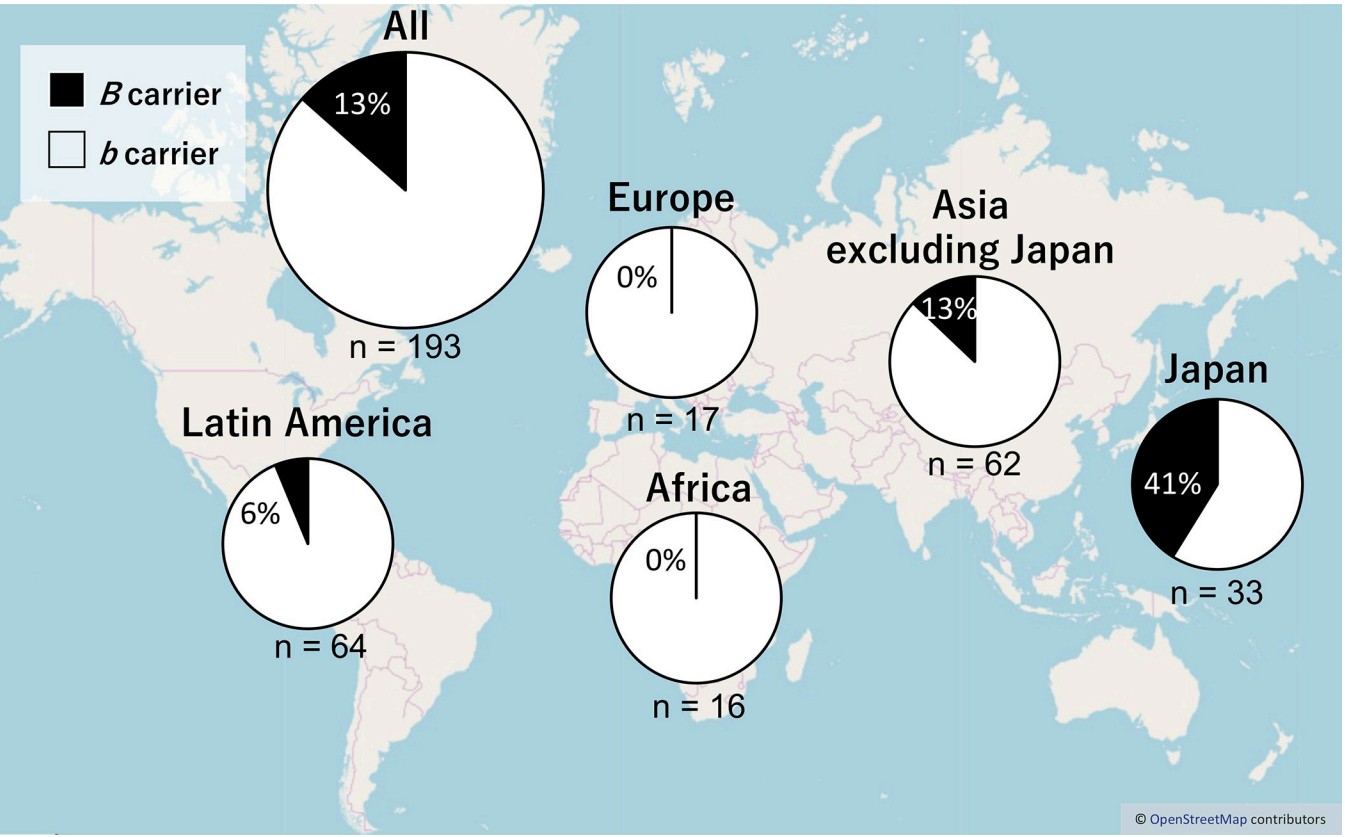

**Fig 1. Distribution and frequency of *B* or *b* allele related to hybrid weakness in *C. annuum*.** The data was based on this and a previous study [11]. The map is based on OpenStreetMap (https://www.openstreetmap.org/copyright).

3). Although PI 213915, PI 177301, and 'Mie' were *B*-carriers, they were not included in the clade with a high percentage of *B*-carriers in the UPGMA and NJ trees.

## Population structure analysis

Population structure analysis was conducted based on the SSR analysis data. As the mean Delta K was maximum at K = 2, optimum K was 2 for population structure analysis (Fig 4). The accessions were divided into two groups (groups A and B) according to population structure (Fig 5). There were *C. pubescens*, *C. frutescens*, *C. baccatum*, *C. chinense*, and six accessions of *C. annuum* from Latin America, including *C. annuum* var. *glabriusculum*, and three accessions of *C. annuum* from Japan in group A, whereas there were 40 *C. annuum* accessions from various regions in group B. *B*-carriers were included in both groups.

## PCoA

PCoA was conducted based on the SSR analysis data to uncover phylogenetic relationships in two dimensions (Fig 6). The first two principal coordinate axes explained ~16.17% of the variation in the genetic distance matrix. The accessions of group A defined in the structural analysis also formed a group in the PCoA. *C. annuum* accessions from Latin America were distributed across a wide area of the graph. Most *B*-carriers were distributed in part of the area indicated as *B*-carrier group, although the area also included some *b*-carriers.

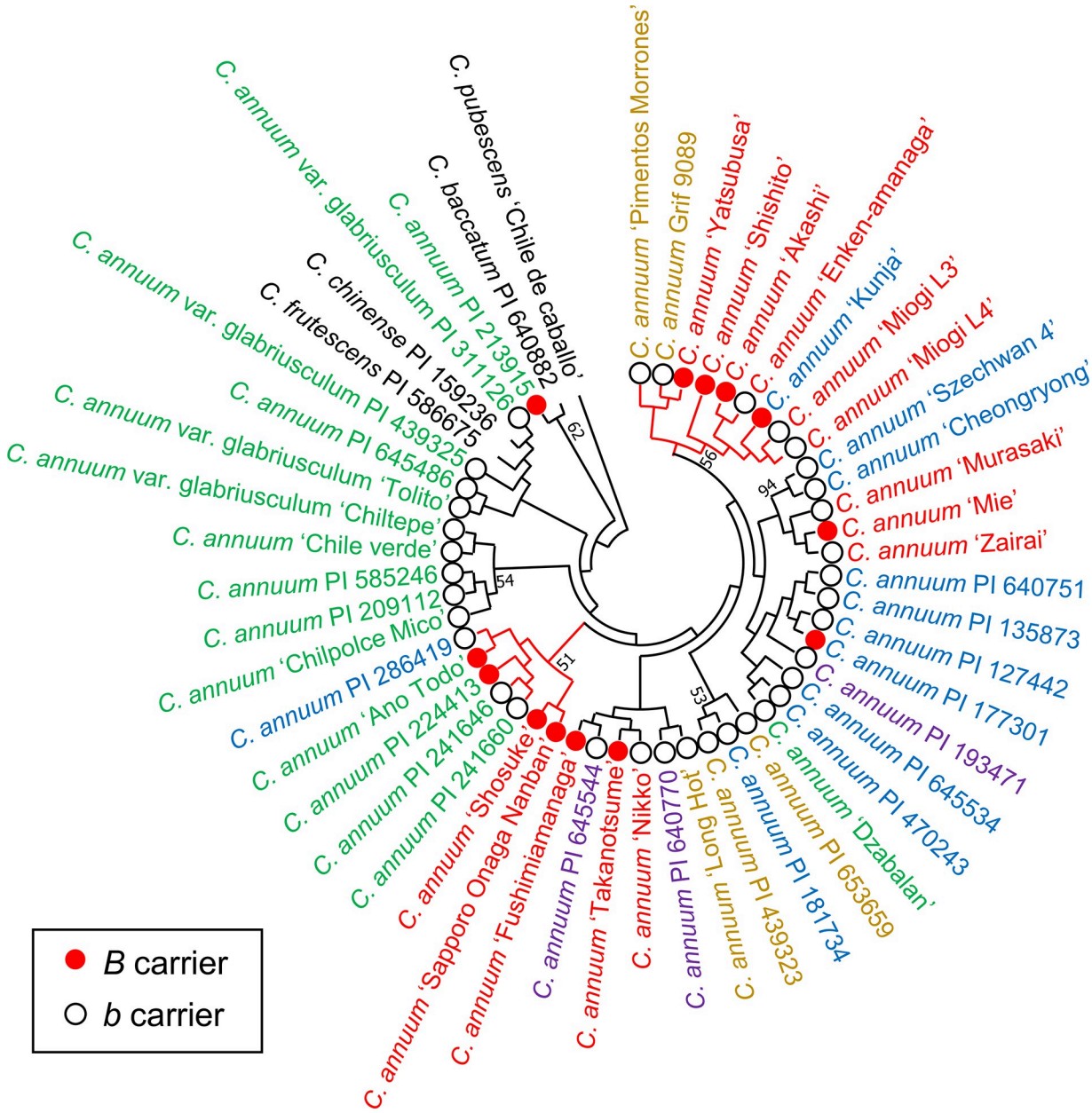

**Fig 2. Phylogenetic UPGMA trees of 48 accessions of *C. annuum* with 4 other species based on 34 SSR markers.** *B* or *b* carriers are indicated as red or white circles, respectively. The branches in the clade with high percentage of *B* carriers are shown in red.

## Discussion

In the present study, SSR analysis was conducted on 52 accessions of domesticated *Capsicum* species, using 34 markers covering all chromosomes. The Ho values were almost zero as heterozygous alleles were not detected for most markers, whereas He values ranged from 0.215 to 0.785 (Table 3). In general, Ho values lower than He values are due to the lack of crossbreeding with other accessions. Low Ho values in the domesticated *Capsicum* species have also been reported in other studies [17, 31]. Therefore, the low Ho value in the present study may suggest

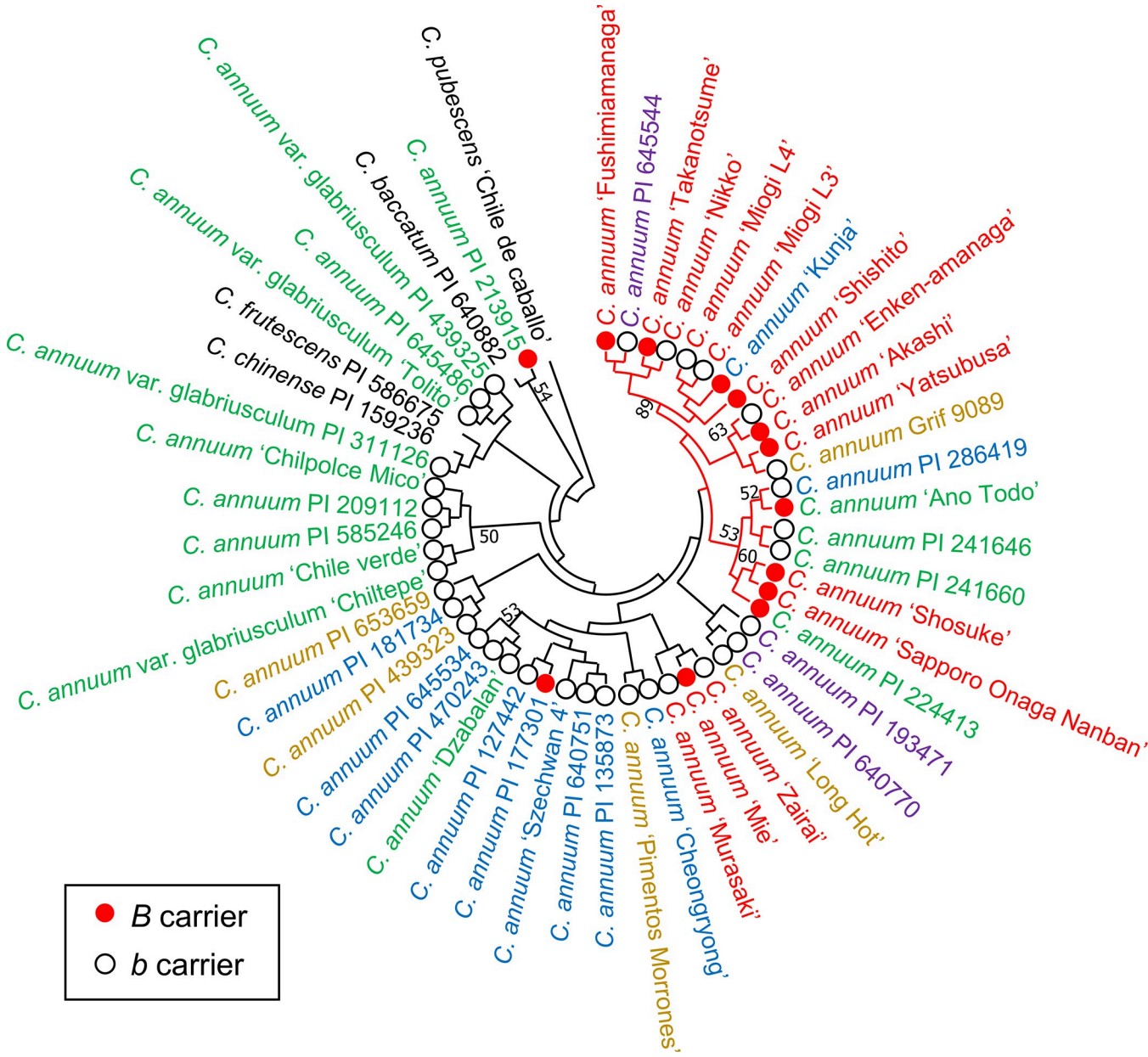

**Fig 3. Phylogenetic NJ trees of 48 accessions of *C. annuum* with 4 other species based on 34 SSR markers.** *B* or *b* carriers are indicated as red or white circles, respectively. The branches in the clade with high percentage of *B* carriers are shown in red.

that the accessions supplied in the present study were fixed at almost all loci as all accessions supplied were domesticated species.

Accessions of *C. annuum* var. *glabriusculum* were closer to *C. chinense* and *C. frutescens* than the other accessions of *C. annuum* (Figs 2, 3, 5, and 6). *C. annuum* var. *glabriusculum* is considered the wild ancestor of *C. annuum*, and *C. chinense* and *C. frutescens* are closely related to *C. annuum* [32, 33]. Therefore, our results show that *C. annuum* evolved via the *C. annuum* var. *glabriusculum* after speciation from the common ancestors of *C. annuum*, *C. chinense*, and *C. frutescens*.

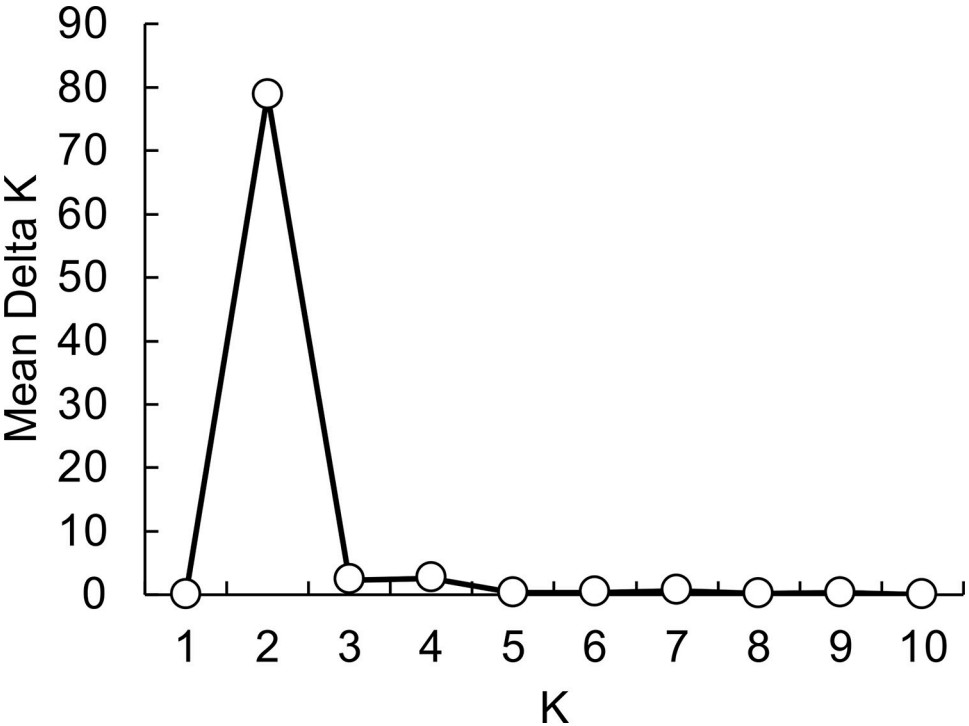

**Fig 4. Determination of optimal K by structure harvester.** K = 2 was considered as the optimal number of populations.

Accessions of *C. annuum* from various countries were investigated using SSR analysis. The accessions of *C. annuum* in Latin America had larger genetic diversity than those in other regions (Fig 6). A diversity center seems to be a region where plants are domesticated from wild species. It is assumed that *C. annuum* differentiated from a common ancestor of related species in Latin America [10, 32]. Moreover, it was found that *C. annuum* was domesticated in Mexico from *C. annuum* var. *glabriusculum* [34]. Therefore, our finding of a large diversity in Latin American lineages supports those of previous studies showing that Latin America is diversity-centric.

There was no particular tendency in the genetic structure by geographic origin, excluding Latin America, in *C. annuum* accessions (Figs 2, 3, 5, and 6). Similarly, a previous study showed that geographic origin had little effect on the genetic structure of *C. annuum* accessions, although *C. annuum* accessions from Turkey and central Europe formed a cluster [17]. Another study examined the phylogenetic relationship of nearly 10,000 lineages around the world, showing some effect of geographic origin on the genetic structure. However, the overlap was significantly higher than segregation in genetic structure among lineages with other regions [19]. Archaeological studies show that *C. annuum* existed more than 6,000 years ago [34, 35], although it is unknown when the species was established. It has only been about 500 years since *C. annuum* was introduced across the world during the Age of Discovery. Therefore, there may be less time for the lineages of *C. annuum* to change their genetic structure by geographic origin.

The accessions of *B*-carrier were in Latin America and Asia, and the percentage of *B*-carriers was particularly high in Japan (Fig 1). Here, we suggest a model showing the process of causal gene acquisition and the global spread of lines having *B* or *b* alleles (Fig 7). The common ancestor of *C. annuum*, *C. chinense*, and *C. frutescens* appears to have had the *aabb* genotype.

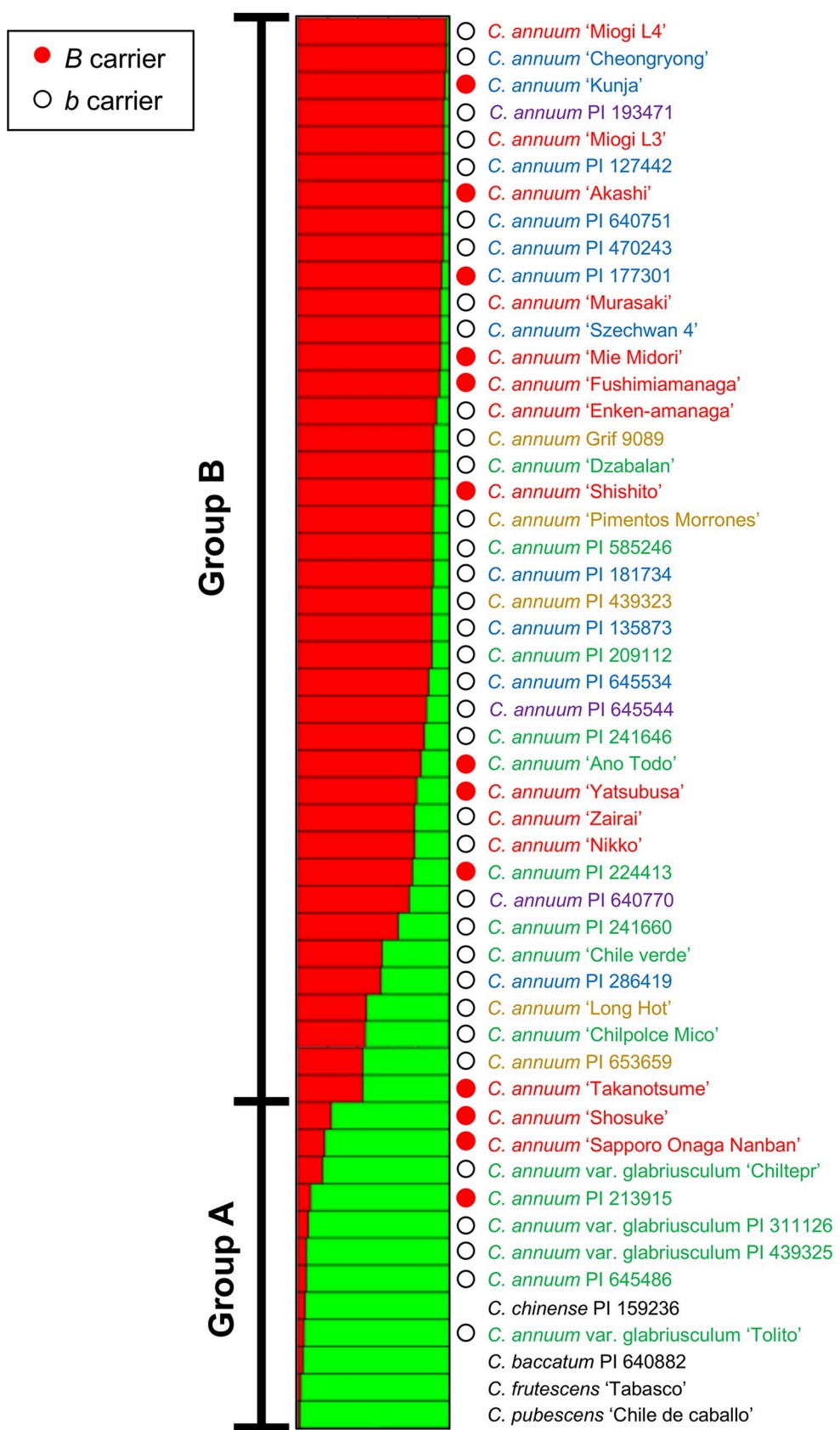

**Fig 5. STRUCTURE analysis in 48 accessions of *C. annuum* and 4 other species based on 34 SSR markers.** *B* or *b* carriers were indicated as respectively red or white circles.

In *C. chinense* and *C. frutescens*, only lines with the *AAbb* genotype have been reported, although no extensive studies have been conducted [11, 13]. All the lines in *C. chinense* and *C. frutescens* might have acquired the *A* gene after speciation from a common ancestor. However, parts of *C. annuum* acquired the *B* gene. No accessions of *C. annuum* var. *glabriusculum*, which is the wild ancestor of *C. annuum*, had the *B* allele in our survey. This suggests that *C. annuum* acquired the *B* allele since domestication or fewer lines have this allele. In the Age of Discovery, *C. annuum* was discovered in Latin America and was introduced globally by trade. Then, most lines brought to the world may be *b*-carriers. However, the lines brought to Asia, particularly Japan, included *B*-carriers. In fact, PI 213915 and PI 224413, the *B*-carriers in Latin America, were genetically close to the accessions in Japan (Fig 5). Therefore, these or similar lines were likely brought to Japan. Therefore, in Asia, *C. annuum* lines would have

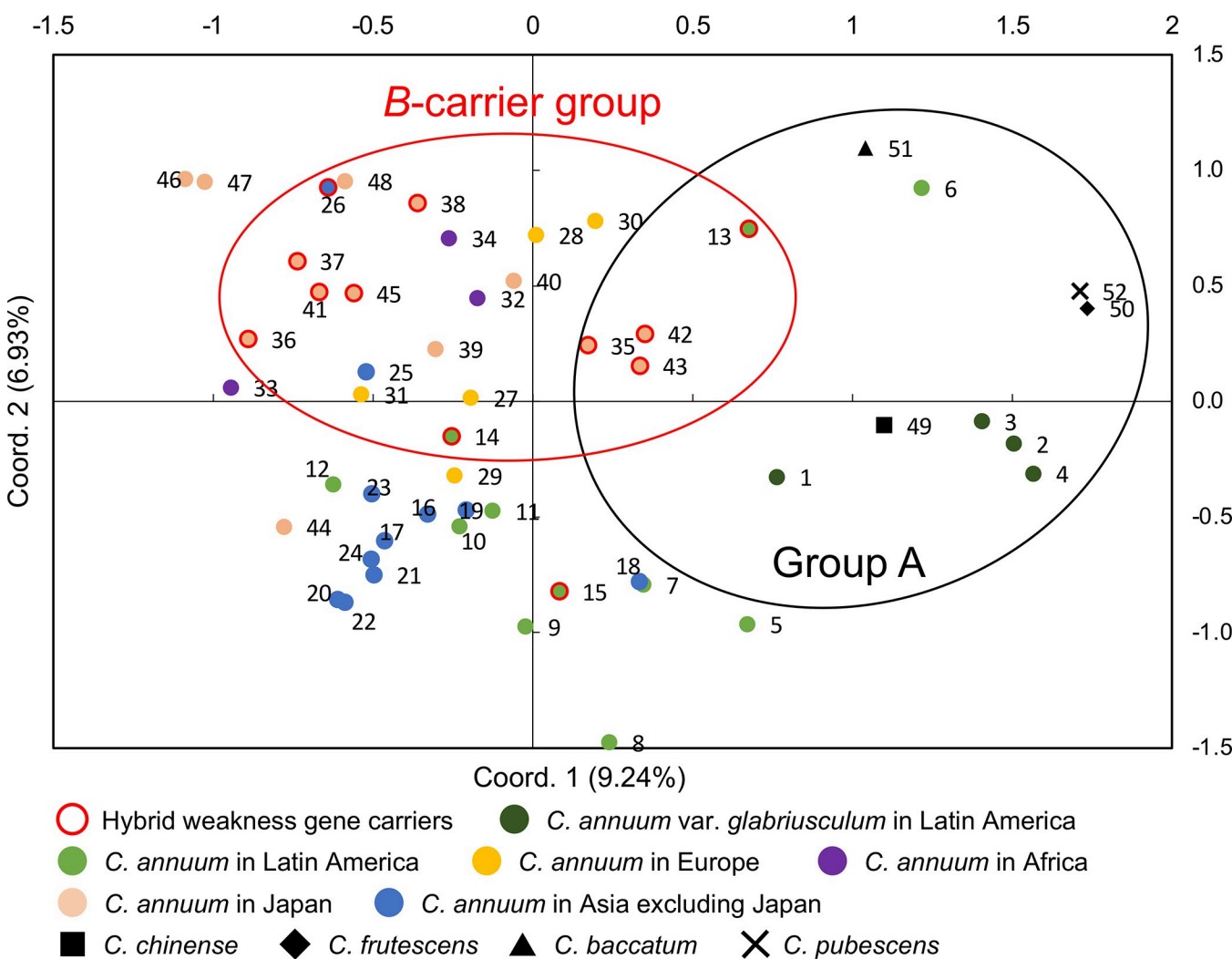

**Fig 6. Principal coordinate analysis in 48 accessions of *C. annuum* and 4 accessions of the other species based on 34 SSR markers.** Group A corresponds to that in Fig 5.

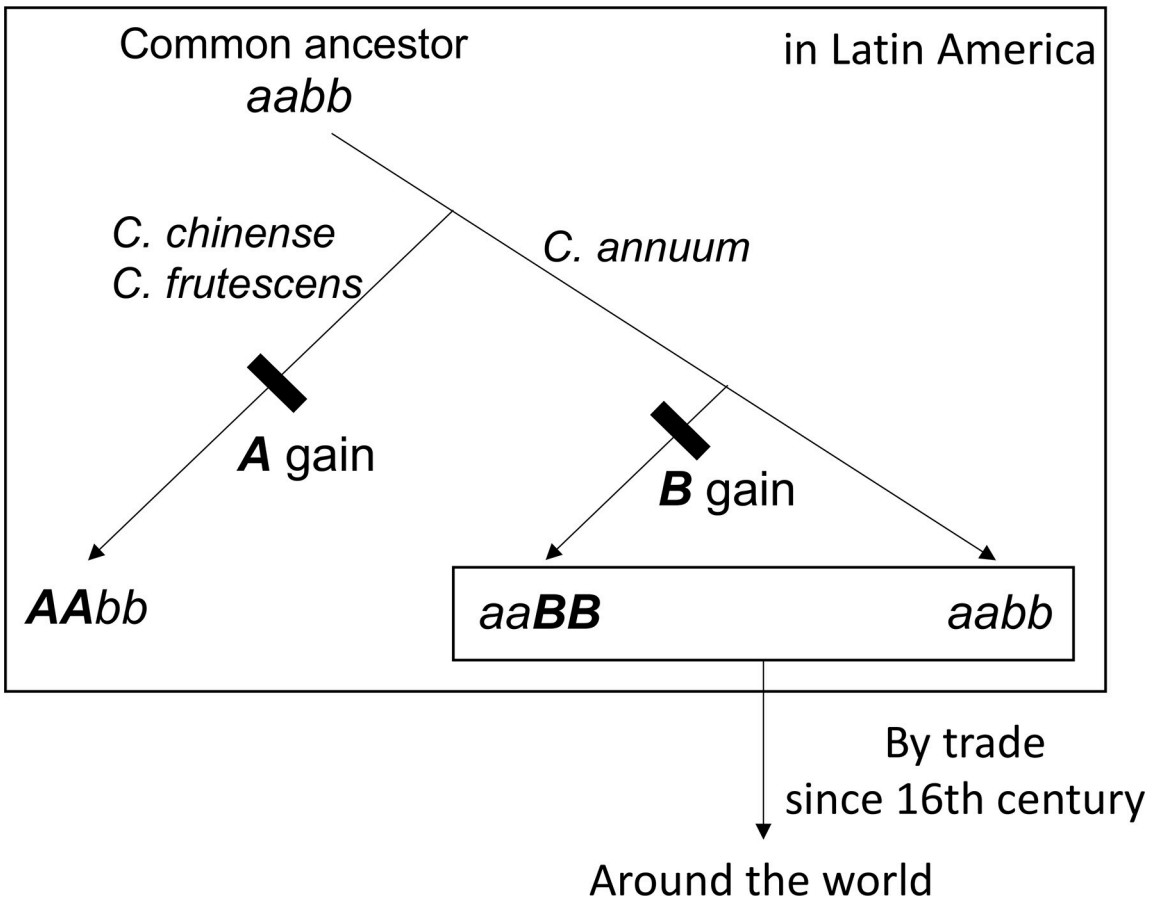

**Fig 7. Model on the process of causal gene acquisition and the spread of the lines of *B*- or *b*-carriers to the world.**

been bred based on the lines of both *B*- and *b*-carriers from Latin America. Perhaps the *B* allele might have functions that show suitable traits for the Asian environment or might be involved in traits preferred by Asians, as *B*-carriers exist at a high ratio in Japanese accessions.

It has been suggested that causal genes for hybrid necrosis may have been used in wheat breeding [36]. In many cases, hybrid weakness or necrosis involves an autoimmune response. Therefore, it is possible that the causal gene contributes to disease resistance [36, 37]. Similarly, *Capsicum* hybrid weakness has been shown to cause an autoimmune response [20], and *B*-allele of *Capsicum* hybrid weakness may have been used in the breeding process.

Our study suggests a history of *B*-allele involvement in *Capsicum* hybrid weakness in acquisition and spread. In the future, identification of causal genes *A* and *B* for hybrid weakness will advance research on speciation in *Capsicum*.

## Supporting information

**S1 Table. Origin and F$_1$ phenotype of *C. annuum* as revealed by test crosses with *C. chinense*.**
(XLSX)

**S2 Table. Information on the primer sequences used in the present study.**
(XLSX)

## Acknowledgments

We are grateful to the National Agriculture and Food Research Organization Genebank (Tsukuba, Japan) and the USDA/ARS *Capsicum* germplasm collection (Griffin, USA) for providing *Capsicum* seeds.

## Author Contributions

**Conceptualization:** Kumpei Shiragaki, Takahiro Tezuka.

**Formal analysis:** Kumpei Shiragaki.

**Funding acquisition:** Takahiro Tezuka.

**Investigation:** Kumpei Shiragaki, Shonosuke Seko.

**Project administration:** Shuji Yokoi, Takahiro Tezuka.

**Supervision:** Shuji Yokoi, Takahiro Tezuka.

**Validation:** Takahiro Tezuka.

**Visualization:** Kumpei Shiragaki.

**Writing – original draft:** Kumpei Shiragaki.

**Writing – review & editing:** Takahiro Tezuka.

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
