## [Decision Letter · Decision Letter 0]

20 May 2022

PONE-D-22-10615Capsicum annuum with causal allele of hybrid weakness is prevalent in AsiaPLOS ONE

Dear Dr. Tezuka,

Thank you for submitting your manuscript to PLOS ONE. After careful consideration, we feel that it has merit but does not fully meet PLOS ONE’s publication criteria as it currently stands. Therefore, we invite you to submit a revised version of the manuscript that addresses the points raised during the review process.

We look forward to receiving your revised manuscript.

Kind regards,

Tzen-Yuh Chiang

Academic Editor

PLOS ONE

Journal Requirements:

2. We note that Figure 1 in your submission contain map/satellite images which may be copyrighted. All PLOS content is published under the Creative Commons Attribution License (CC BY 4.0), which means that the manuscript, images, and Supporting Information files will be freely available online, and any third party is permitted to access, download, copy, distribute, and use these materials in any way, even commercially, with proper attribution. For these reasons, we cannot publish previously copyrighted maps or satellite images created using proprietary data, such as Google software (Google Maps, Street View, and Earth). For more information, see our copyright guidelines: http://journals.plos.org/plosone/s/licenses-and-copyright.

a) You may seek permission from the original copyright holder of Figure 1 to publish the content specifically under the CC BY 4.0 license.  

Reviewers' comments:

Reviewer's Responses to Questions

**Comments to the Author**

1. Is the manuscript technically sound, and do the data support the conclusions?

Reviewer #1: Yes

Reviewer #2: Yes

Reviewer #3: Yes

2. Has the statistical analysis been performed appropriately and rigorously? 

Reviewer #1: Yes

Reviewer #2: Yes

Reviewer #3: Yes

3. Have the authors made all data underlying the findings in their manuscript fully available?

Reviewer #1: Yes

Reviewer #2: Yes

Reviewer #3: Yes

4. Is the manuscript presented in an intelligible fashion and written in standard English?

Reviewer #1: Yes

Reviewer #2: Yes

Reviewer #3: Yes

5. Review Comments to the Author

Reviewer #1: The manuscript is well written and I would like to congratulate the authors to the interesting manuscript and would like to recommend it for publication. the findings reported in the manuscript 'Capsicum annuum with causal allele of hybrid weakness is prevalent in Asia' will have great impact in the field of Capsicum sp. breeding.

Reviewer #2: In this manuscript, the authors reported their study of the presence of B or b allele for hybrid weakness in Capsicum annuum accessions from different countries. The work also inculded previous results which had been published in Japanese. Crossing between accessions with different genotypes were performed and SSR markers were analyzed. The experiments were nicely performed, and data were convincing. The manuscript is written in a clean and clear manner, and is easy to follow. I have only a few tiny suggestions.

1. Line 133, "tris" should be "Tris".

2. Line 142, "Mega" should be "MEGA".

3. Lines 84-86, I would suggest to revise it as "We used 94 C. annuum accessions (including 16 C. annuum var. glabriusculum accessions, Table S1), one C. chinese accession...". In this way, readers would not try to find information of other species from Table S1.

4. Line 107, ", Tokyo, Japan" can be removed. It has been introduced for Sakata Seed Co. before (line 88).

Reviewer #3: Reproductive isolation, including hybrid weakness, plays an important role in the formation of species.In this manuscript, authors surveyed whether 94 C. annuum accessions had B or b alleles by crossing with C. chinense having the A allele. Of the 94 C. annuum accessions, five had the B allele, three of which were native to Latin America and two were native to Asia. When combined with previous studies, the percentage of B carriers was 41% in Japan, 13% in Asia excluding Japan, 6% in Latin America, and 0% in Europe and Africa. In addition, 48 accessions of C. annuum from various countries were subjected to SSR analysis. Clades with high percentages of B -carriers were formed in the phylogenetic trees. In the principal coordinate analysis, most B -carriers were localized in a single group, although the group also included b -carriers. Based on these results, we presumed that the B allele was acquired in some C. annuum lines in Latin America, and B -carriers were introduced to the world during the Age of Discovery, as along with the b -carriers.

The manuscript has been well written. A good result about the B allele from C. annuum has been identified. It would be useful for the relative people to know.

(1)Line 61- 62: Capsicum migrated to Europe and Africa from Latin America and to Southeast Asia during the 17–18th centuries from Peru.

From my view, “pepper was introduced to Southeast Asia during the 16th centuries probably by Portuguese .”

For evidence:

In 1542, the Portuguese brought pepper to Goa, India, and pepper began to spread in South Asia. Then it spread to Malacca and spread in Southeast Asia. At the same time, the Spanish brought pepper to Luzon, the Philippines, which is another transmission point of pepper in Southeast Asia (watt, 2014).

It was first recorded in Japanese literature that chili was introduced in 1552. Balthazar Gago, a Portuguese missionary, gave it as a gift to daiyo Town, which was then known as Toyoda and Toyoda of Kyushu Island, 39 years earlier than China (Yamamoto, 2018).

During the compilation of Korean "history of Korea", it was introduced into the Korean "history of Korea" in 1602.

The earliest record of pepper in China can be found in Zunsheng eight notes by Gao Lian of the Ming Dynasty (1591): "pepper is born, white flowers, and the fruit looks like bald pen head. It tastes spicy and red, which is very impressive."

Reference:

Watt G. 2014. A dictionary of the economic products of India. Landon: Cambridge University Press：197

Yoshimoto Yamamoto (translated by Chen Xian-ruo ). 2018. World history of pepper: a journey across Europe, Asia, and Africa, a hot table revolution. Taiwan:Marco Polo Press：256

(2)Line 114-, “DNA extraction” Whose DNA will be extracted should be mentioned.

6. PLOS authors have the option to publish the peer review history of their article (what does this mean?). If published, this will include your full peer review and any attached files.

Reviewer #1: **Yes: **Julia Meitz-Hopkins, PhD

Reviewer #2: No

Reviewer #3: No

---

## [Author Response · Author response to Decision Letter 0]

10 Jun 2022

Dear Dr. Tzen-Yuh Chiang,

We would like to thank reviewers for their valuable comments. We revised our manuscript according to their comments. Below, we have included point-by-point responses (black text) to your and reviewers’ comments (blue text).

We checked the format of the paper.

2. We note that Figure 1 in your submission contain map/satellite images which may be copyrighted. All PLOS content is published under the Creative Commons Attribution License (CC BY 4.0), which means that the manuscript, images, and Supporting Information files will be freely available online, and any third party is permitted to access, download, copy, distribute, and use these materials in any way, even commercially, with proper attribution. For these reasons, we cannot publish previously copyrighted maps or satellite images created using proprietary data, such as Google software (Google Maps, Street View, and Earth).

The map in Figure 1 was obtained from OpenStreetMap under Open Data Commons Open Database License. Therefore, we can use the map under the Creative Commons Attribution License. We added a sentence and a link of copyright (lines 172-173).

Reviewer #1: 

The manuscript is well written and I would like to congratulate the authors to the interesting manuscript and would like to recommend it for publication. the findings reported in the manuscript 'Capsicum annuum with causal allele of hybrid weakness is prevalent in Asia' will have great impact in the field of Capsicum sp. breeding.

Thank you for reviewing and evaluating our paper.

Reviewer #2: 

In this manuscript, the authors reported their study of the presence of B or b allele for hybrid weakness in Capsicum annuum accessions from different countries. The work also inculded previous results which had been published in Japanese. Crossing between accessions with different genotypes were performed and SSR markers were analyzed. The experiments were nicely performed, and data were convincing. The manuscript is written in a clean and clear manner, and is easy to follow. I have only a few tiny suggestions.

Thank you for pointing out some mistakes. We revised each point as follows.

1. Line 133, "tris" should be "Tris".

We revised that as you pointed out (line 134).

2. Line 142, "Mega" should be "MEGA".

We revised that as you pointed out (line 143).

3. Lines 84-86, I would suggest to revise it as "We used 94 C. annuum accessions (including 16 C. annuum var. glabriusculum accessions, Table S1), one C. chinese accession...". In this way, readers would not try to find information of other species from Table S1.

Thank you for suggesting a very nice revision. We revised the sentence as you suggested (line 84-86).

4. Line 107, ", Tokyo, Japan" can be removed. It has been introduced for Sakata Seed Co. before (line 88).

We revised that as you pointed out (line 107).

Reviewer #3: Reproductive isolation, including hybrid weakness, plays an important role in the formation of species.In this manuscript, authors surveyed whether 94 C. annuum accessions had B or b alleles by crossing with C. chinense having the A allele. Of the 94 C. annuum accessions, five had the B allele, three of which were native to Latin America and two were native to Asia. When combined with previous studies, the percentage of B carriers was 41% in Japan, 13% in Asia excluding Japan, 6% in Latin America, and 0% in Europe and Africa. In addition, 48 accessions of C. annuum from various countries were subjected to SSR analysis. Clades with high percentages of B -carriers were formed in the phylogenetic trees. In the principal coordinate analysis, most B -carriers were localized in a single group, although the group also included b -carriers. Based on these results, we presumed that the B allele was acquired in some C. annuum lines in Latin America, and B -carriers were introduced to the world during the Age of Discovery, as along with the b -carriers.

The manuscript has been well written. A good result about the B allele from C. annuum has been identified. It would be useful for the relative people to know.

Thank you for reviewing our manuscript. According to your advice, we revised the manuscript as mentioned below.

(1)Line 61- 62: Capsicum migrated to Europe and Africa from Latin America and to Southeast Asia during the 17–18th centuries from Peru.

From my view, “pepper was introduced to Southeast Asia during the 16th centuries probably by Portuguese .”

For evidence:

In 1542, the Portuguese brought pepper to Goa, India, and pepper began to spread in South Asia. Then it spread to Malacca and spread in Southeast Asia. At the same time, the Spanish brought pepper to Luzon, the Philippines, which is another transmission point of pepper in Southeast Asia (watt, 2014).

It was first recorded in Japanese literature that chili was introduced in 1552. Balthazar Gago, a Portuguese missionary, gave it as a gift to daiyo Town, which was then known as Toyoda and Toyoda of Kyushu Island, 39 years earlier than China (Yamamoto, 2018).

During the compilation of Korean "history of Korea", it was introduced into the Korean "history of Korea" in 1602.

The earliest record of pepper in China can be found in Zunsheng eight notes by Gao Lian of the Ming Dynasty (1591): "pepper is born, white flowers, and the fruit looks like bald pen head. It tastes spicy and red, which is very impressive."

Reference:

Watt G. 2014. A dictionary of the economic products of India. Landon: Cambridge University Press：197

Yoshimoto Yamamoto (translated by Chen Xian-ruo ). 2018. World history of pepper: a journey across Europe, Asia, and Africa, a hot table revolution. Taiwan:Marco Polo Press：256

Thank you for pointing this out. Andrew (1995) also mentioned that it was introduced to Asia in the 16th century. We corrected the centuries when pepper was introduced to Asia (line 61-62). 

Andrews J. Peppers: the domesticated Capsicums. University of Texas Press; 1995

(2)Line 114-, “DNA extraction” Whose DNA will be extracted should be mentioned.

We added the information on whose DNA (line 115).

---

## [Decision Letter · Decision Letter 1]

24 Jun 2022

Capsicum annuum with causal allele of hybrid weakness is prevalent in Asia

PONE-D-22-10615R1

Dear Dr. Tezuka,

We’re pleased to inform you that your manuscript has been judged scientifically suitable for publication and will be formally accepted for publication once it meets all outstanding technical requirements.

Kind regards,

Tzen-Yuh Chiang

Academic Editor

PLOS ONE

Additional Editor Comments (optional):

Reviewers' comments:

Reviewer's Responses to Questions

**Comments to the Author**

1. If the authors have adequately addressed your comments raised in a previous round of review and you feel that this manuscript is now acceptable for publication, you may indicate that here to bypass the “Comments to the Author” section, enter your conflict of interest statement in the “Confidential to Editor” section, and submit your "Accept" recommendation.

Reviewer #1: All comments have been addressed

Reviewer #2: All comments have been addressed

Reviewer #3: All comments have been addressed

2. Is the manuscript technically sound, and do the data support the conclusions?

Reviewer #1: No

Reviewer #2: Yes

Reviewer #3: Yes

3. Has the statistical analysis been performed appropriately and rigorously? 

Reviewer #1: Yes

Reviewer #2: Yes

Reviewer #3: Yes

4. Have the authors made all data underlying the findings in their manuscript fully available?

Reviewer #1: Yes

Reviewer #2: Yes

Reviewer #3: Yes

5. Is the manuscript presented in an intelligible fashion and written in standard English?

Reviewer #1: Yes

Reviewer #2: Yes

Reviewer #3: Yes

6. Review Comments to the Author

Reviewer #1: All comments have been addressed by the author. As mentioned in the initial review, the manuscript is well written and I would like to congratulate the authors to the interesting manuscript and would like to recommend it for publication. the findings reported in the manuscript 'Capsicum annuum with causal allele of hybrid weakness is prevalent in Asia' will have great impact in the field of Capsicum sp. breeding.

Reviewer #2: The authors have made revisions according to all comments from the reviewers. I think it is ready to be accepted.

Reviewer #3: Hybrid weakness plays an important role in the formation of species. In the manuscript, authors surveyed whether 94 C. annuum accessions had B or b alleles by crossing with C. chinense having the A allele. Of the 94 C. annuum accessions, five had the B allele, three of which were native to Latin America and two were native to Asia. Based on these results, the B allele was acquired in some C. annuum lines in Latin America, and B -carriers were introduced to the world during the Age of Discovery, as along with the b -carriers has beed concluded.

The comments from me have been well addressed. I have no more comments and suggestion on the manscript.

I agree to accept it for publication.

7. PLOS authors have the option to publish the peer review history of their article (what does this mean?). If published, this will include your full peer review and any attached files.

Reviewer #1: **Yes: **Julia Meitz-Hopkins

Reviewer #2: No

Reviewer #3: No

---

## [Editor Report · Acceptance letter]

28 Jun 2022

PONE-D-22-10615R1 

*Capsicum annuum* with causal allele of hybrid weakness is prevalent in Asia 

Dear Dr. Tezuka:

I'm pleased to inform you that your manuscript has been deemed suitable for publication in PLOS ONE. Congratulations! Your manuscript is now with our production department. 

Kind regards, 

on behalf of

Dr. Tzen-Yuh Chiang 

Academic Editor

PLOS ONE